# The Impact of Return-to-Field and Targeted Trap-Neuter-Return on Feline Intake and Euthanasia at a Municipal Animal Shelter in Jefferson County, Kentucky

**DOI:** 10.3390/ani10081395

**Published:** 2020-08-11

**Authors:** Daniel D. Spehar, Peter J. Wolf

**Affiliations:** 1Independent Researcher, 4758 Ridge Road, #409, Cleveland, OH 44144, USA; danspehar9@gmail.com; 2Best Friends Animal Society, 5001 Angel Canyon Road, Kanab, UT 84741, USA

**Keywords:** return-to-field (RTF), trap-neuter-return (TNR), targeted TNR, community cats, stray cats, feral cats, urban free-roaming cats, animal sheltering, feline intake, feline euthanasia

## Abstract

**Simple Summary:**

Nearly three-quarters of a million cats and dogs were euthanized at animal shelters in the United States in 2018. This total represents a decline of more than 90% in such deaths at USA shelters since the early 1970s. The majority of animals euthanized in shelters today are free-roaming feral and stray cats. Two new approaches to the management of free-roaming cats—return-to-field (RTF) and targeted trap-neuter-return (TNR)—have grown in use over the past decade and have recently been associated with significant reductions in shelter euthanasia and intake. RTF programs are similar to TNR programs in that they entail the sterilization, vaccination, and return of cats; however, RTF programs are shelter based rather than community based. RTF programs provide live outcomes for community cats otherwise at high risk of euthanasia after shelter admission. The purpose of the present study was to examine changes in feline euthanasia and intake, as well as additional shelter metrics, at a municipal animal shelter in Jefferson County, KY, USA, after an RTF program was added to an ongoing community-based TNR program. The euthanasia of cats at Louisville Metro Animal Services (LMAS) declined by 94.1% and feline admissions dropped by 42.8% after a combined total of 24,697 cats were trapped, sterilized, vaccinated and returned over an eight-year period. The results of the present study are consistent with previous research and illustrate the amenability of RTF and TNR programs to customization based upon the needs and resources of a given community.

**Abstract:**

The number of cats and dogs impounded and euthanized at animal shelters in the USA has declined dramatically in recent decades. The Humane Society of the United States reported that in 1973 an estimated 13.5 million cats and dogs were euthanized nationwide; according to Best Friends Animal Society, in 2018 that number had been reduced to approximately 733,000. A disproportionate number of animals euthanized at shelters today are free-roaming feral and stray cats, who most often face euthanasia due to their temperament or a lack of shelter space. Over the past decade, two new management tactics—return-to-field (RTF) and targeted trap-neuter-return (TNR)—have exhibited the capacity to contribute to significant reductions in feline euthanasia and intake. The present study examines changes in feline euthanasia and intake, as well as impacts on additional metrics, at a municipal animal shelter in Jefferson County, KY, USA, after an RTF program was added to an ongoing community-based TNR program. A combined total of 24,697 cats were trapped, sterilized, vaccinated, and returned over 8 years as part of the concurrent RTF and TNR programs. Feline euthanasia at Louisville Metro Animal Services (LMAS) declined by 94.1% and feline intake dropped by 42.8%; the live-release rate (LRR) increased by 147.6% due primarily to reductions in both intake and euthanasia. The results of the present study corroborate prior research on the effectiveness of combining RTF and TNR and exemplify the flexibility available to communities in configuring such programs to align with their particular needs and resources.

## 1. Introduction

As recently as the 1970s, an estimated 90% of the cats and dogs entering USA animal shelters were euthanized [1,2]. Based upon its nationwide survey of shelters in 1973, The Humane Society of the United estimated that 13.5 million cats and dogs were being euthanized annually at such facilities, which amounted to approximately 64 animals per 1000 USA residents [3]. In 2018, an estimated total of 5.3 million cats and dogs were admitted to USA shelters, 733,000 of whom were euthanized [4]**—**approximately 2.2 per 1000 USA residents [5]. Widespread companion animal sterilization campaigns that began in the early 1970s are generally credited with these dramatic declines in the number of cats and dogs admitted to and euthanized at USA animal shelters [1,3,6].

Most of the cats euthanized at USA shelters today are in good general health upon admission, but are nevertheless euthanized due to their temperament or because of a lack of shelter space [7,8,9,10]; many are free-roaming feral or stray cats (henceforth referred to as community cats) who are considered poor candidates for adoption. Over the past decade, an alternative to the practice of euthanizing unadoptable community cats has emerged. A growing number of shelters are instituting shelter-based neuter-return programs, commonly known as return-to-field (RTF), but sometimes referred to as shelter-neuter-return (SNR). These programs are intended to provide live outcomes for cats designated as “strays” upon shelter admission (either brought by residents or impounded by field services staff) [11]. After enrollment, cats in RTF programs are sterilized, vaccinated, and ear tipped before being returned to their locations of origin [12]. RTF programs, such as community-based TNR programs, are implemented with the same two-fold objective of reducing (1) the number of cats who, either due to temperament or lack of space, would otherwise likely be euthanized, and (2) community cat populations. A number of RTF programs, particularly those implemented in combination with high-intensity, targeted TNR, have been effective at reducing both feline intake and euthanasia at municipal shelters [11,13,14].

The present study examines changes in feline euthanasia and intake, and other metrics, over an eight-year period at a municipal shelter in Jefferson County, Kentucky, after implementation of a two-pronged initiative that paired an RTF program with an ongoing community-based TNR program. Unlike the integrated RTF/TNR programs previously reported in the peer-reviewed literature, which were jointly conducted by shelter personnel and the staff of a national animal welfare organization [11,13], the Jefferson County programs were operated on a concurrent, yet independent basis—the RTF program operated by municipal shelter staff and the TNR program, including the sustained targeting of specific zip codes, conducted by a private community cat advocacy organization. Because the coexistent RTF and TNR programs in Jefferson County were operated separately, rather than integrated, as had been carried out elsewhere, and lacked some of the resources available to previously documented programs (e.g., TNR program staff working directly with shelter staff), it was unclear whether similar reductions in euthanasia and intake could be achieved.

### 1.1. Key Terminology

The following terms and acronyms, which might be unfamiliar to some readers, are used throughout this paper (adapted from [10]).

Feral cats: Generally used to describe unowned, free-roaming cats, especially those who are not socialized to humans either because they have never had human contact or because they have, over time, reverted to a “wilder” state in the absence of such human contact.Stray cats: Generally used to describe owned cats (i.e., pets) who have become lost or displaced from their homes or property. Shelters generally use the term more broadly, considering cats admitted to be “strays” if they have no known owner and lack discernible identification (i.e., collar and tags, or traceable microchip).Community cats: Broadly used to describe unowned, free-roaming cats regardless of their sociability. The term is becoming more popular in part because of its implicit acknowledgment that cats are a commensal species and are highly valued by many residents.Trap-neuter-return (TNR): A non-lethal method for managing community cats whereby cats are humanely trapped, surgically sterilized, and returned to their original location. TNR programs often vaccinate cats against rabies as well (Figure 1).Return-to-field (RTF): Essentially TNR for cats brought into an animal shelter, either by residents or a shelter’s field services staff, as “strays” lacking identification (i.e., collar/tags or traceable microchip). Rather than house these cats only to kill most of them following the designated “stray hold” period, shelters sterilize, vaccinate, and return them to the outdoor location where the cats were living. Also sometimes called shelter-neuter-return, or SNR (Figure 1).Community Cat Program (CCP): The name given to shelter-based programs in which RTF and TNR efforts were carefully integrated in an attempt to maximize lifesaving and population reduction.Sterilization: Various terms are used to indicate the surgical sterilization of cats (e.g., spay, neuter). Here, we are using the term sterilization broadly to indicate the removal of the reproductive organs in both male (testicles) and female (ovaries) cats.Alley Cat Advocates (ACA): A private non-profit organization located in Louisville, KY, that provides no-cost TNR services to residents of the Louisville Metro area.Louisville Metro Animal Services (LMAS): The municipal animal shelter serving residents of the Louisville Metro area.Kentucky Humane Society (KHS): A private non-profit organization located in Louisville, KY, that provides a range of services (e.g., low-cost sterilization of pets and adoption of pets) mostly to residents of the Louisville Metro area.Live-release Rate (LRR): A metric commonly used to describe animal shelter outcomes, defined as live outcomes divided by all outcomes except owner-requested euthanasia and animals who died or were lost while in the shelter’s care, represented as a percentage.Return to Owner (RTO): The ratio of animals reunited with their owners to total shelter intake, represented as a percentage.Length of Stay (LOS): The amount of time (usually expressed in days, as an average) between an animal’s admission to a shelter and their eventual outcome.Euthanasia Rate: The ratio of the number of cats euthanized for reasons other than owner request to the total number of feline intakes over a given period of time, represented as a percentage.

### 1.2. Site Description and Background Information

Jefferson County is located in North Central Kentucky, USA, situated along the Ohio River at Indiana’s southern edge. Louisville is Kentucky’s largest city and serves as the seat of Jefferson County. A 2003 city–county merger resulted in the formation of the Louisville/Jefferson County Metro Government (a.k.a. Louisville Metro, KY, USA) [15]. Jefferson County has an estimated total population of 770,517 [16]. Louisville Metro Animal Services (LMAS), which serves all of Jefferson County, operated a 28,000 square-foot (~2600 square meter) animal shelter (open-admission for the county’s stray animals and managed-intake for owned animals beginning in 2017) that received and housed animals along with a separate off-site building for adoptions until late in 2019, when the building used to admit and hold animals was replaced by a new 33,000 square-foot (~3065 square meter) animal shelter located on the same campus as the adoption center. Facilities with an open-admission policy generally accept any animal in need, including those with little chance of being rehomed due to issues of age, health, or temperament, although admission restrictions based upon municipal borders or defined hours of animal intake may exist [17,18]. Under a managed-admission policy, the intake of owner-surrendered animals is scheduled by appointment in order to match inflow into the shelter with available space and resources [18].

Alley Cat Advocates (ACA), a private non-profit organization dedicated to providing no-cost TNR services to residents of Louisville Metro, and to surrounding areas as resources allowed, began operations in 1999. During ACA’s first decade of operation, an average of more than 1600 cats were sterilized per year as part of their TNR program. Surgeries were performed either by local veterinarians who volunteered their services as part of large-scale monthly sterilization events (a.k.a. the “Big Fix” program) or at the Kentucky Humane Society’s high-quality, high-volume spay-neuter clinic (a.k.a. the “Quick Fix” program, which was operated out of a variety of privately run clinics before being moved to the Kentucky Humane Society (KHS) in 2011). In 2010, after obtaining a two-year grant from PetSmart Charities, Inc., ACA initiated a program of intensive TNR, targeting the zip code (40215) from which the most cats were being brought to the municipal shelter by residents and field services staff (a.k.a. animal control officers, or ACOs).

The targeted area is a low-income urban community of nearly 22,000 residents, and home to the LMAS shelter until 2019. Median household income in 2012 for zip code 40215 was $26,628 compared to $46,701 for Jefferson County as a whole [19]; population density per square mile in 2012 for the target zip code was 5853 [20] versus 1949 for the county in its entirety [21]. The objective of the program was to trap, sterilize, vaccinate, and return 2000 community cats over a 24-month period. However, it became apparent to ACA staff as the program progressed that there were not enough community cats in the target area to meet the 2000 surgery goal in the allotted amount of time, so four additional zip codes were added to the program. Upon program completion, 1200 community cats had been trapped, sterilized, vaccinated and returned to zip code 40215. Over the same two-year period, stray feline intake at LMAS originating from zip code 40215 declined by 51% (from 1119 in 2009 to 550 in 2011) compared to a decline of 20% (4016 in 2009 to 3206 in 2011) experienced in the remainder of the shelter’s service area [22]. In addition, the total number of cats euthanized at LMAS declined by 47% (from 4970 to 2626) over the same period (no zip code-specific data were available).

These results, along with advocacy from ACA, were the impetus behind municipal leaders enacting in 2012 an ordinance sanctioning the use of TNR as the official method by which community cats would be managed in the Louisville Metro area. Key provisions of the ordinance include the exclusion of “unowned” community cats from at-large prohibitions, licensing requirements, and the mandatory 5-day holding period for new admissions at LMAS. The new ordinance also provided that the municipal government fund a TNR program for community cats to be operated by LMAS, or its designee, to include “sterilization, vaccination, ear tipping for easy identification, and any providing of other necessary medical care for community cats” [23]. Soon after the enactment of the new ordinance, a program designed for community cats based upon the RTF concept was initiated at LMAS, while ACA continued to provide TNR services to residents of Jefferson County (Figure 2). The LMAS community cat program was structured so that all healthy community cats (including those deemed healthy after receiving medical care) were returned to their location of origin after receiving sterilization surgery; fluids (as appropriate); rhinotracheitis/calciviris/panleukopenia (FVRCP) and three-year rabies vaccinations; an ear tip; deworming; and treatment for fleas, ear mites, and other medical conditions, as warranted. After recovery, cats were returned to trapping sites by field services staff. Cats were not relocated unless their home environments were determined to be too dangerous for a safe return, which occurred only rarely. When such situations arose, beginning in 2018, cats were enrolled in an in-house working cat program; prior to 2018, such cats were transferred to a working cat program operated by the KHS. Working cat programs provide an alternative to euthanasia for community cats who are unable to be returned to their location of origin, but who are deemed unadoptable due to temperament by allowing the cats to be relocated, most often to a place of business (e.g., brewery or garden center), where after a period of acclimation the cats are allowed to roam freely in order to deter the presence of rodents [24,25]. Fewer than 20 cats per year were typically enrolled in either working cat program. Calls made by residents to LMAS or other municipal offices about community cats have been referred, by contract, to ACA since 2018. ACA provides TNR and community cat complaint mitigation services for county residents.

## 2. Materials and Methods

### 2.1. Data Collection

For the purposes of the present study, all data pertaining to LMAS (shelter metrics and RTF program results) and ACA (TNR program results) were obtained directly from the respective entity, unless otherwise noted.

Upon admission, information about cats entering LMAS was entered by shelter staff into Chameleon software (Littleton, CO, USA). Community cats in general good health and of sufficient size to safely undergo surgery—whether brought to the shelter by residents or LMAS field services staff—were enrolled in the RTF program, whereby they were sterilized, vaccinated, and returned to their locations of capture. For the purposes of this study, community cats who had been sterilized prior to their arrival at the shelter were not counted as part of the RTF program. Such cats were most often brought in for medical treatment, and were returned to their locations of origin after appropriate ministration.

Cats known or believed to be owned and those deemed too unhealthy for return even after treatment were not enrolled in the RTF program. Beginning in 2014, cats assigned to the LMAS RTF program were tracked by zip code.

Information about cats enrolled in the ACA TNR program, was entered by staff onto Microsoft Excel (Redmond, WA, USA) spreadsheets. ACA tracked cats by sex, age category (over or under 6 months), already sterilized, pregnant (gravid), in-heat (estrus), cryptorchidism, and zip code. The number of sterilization surgeries performed and the number of cats who died in care were also tracked. Outcomes for cats after their return to trapping sites were not tracked as part of either the LMAS RTF program or the ACA TNR program.

Both the ACA and LMAS programs sterilized community cats originating from throughout Jefferson County’s 39 zip codes (for which demographic data are available). ACA tracked cats originating from zip codes both inside and out of Jefferson County for the duration of the program. LMAS tracked cats by individual zip code, but these records were available going back to only 2014. Ten “target” zip codes (40208, 40209, 40213, 40214, 40215, 40216, 40217, 40219, 40258, and 40272) were the source of the majority of cats enrolled in both the RTF and targeted TNR programs. The target area is made up of mostly urban communities, including zip code 40215, where ACA conducted its 2010–11 targeted TNR campaign, mentioned previously. The intensity of TNR activity conducted in particular zip codes was dictated by two factors: (1) the volume of requests received from residents (by LMAS and ACA) for service, and (2) by the number of additional unsterilized community cats residing at or near TNR program trapping/return sites, as reported to ACA during routine follow-up calls to colony caregivers. Follow-up contacts took place during the autumn and winter months, when fewer kitten births generally occur [26] and the number of resident requests for service received by ACA declined. During these months, follow-up contacts with colony caregivers typically were made within 30 days of cats being returned; however, due to a heightened workload, follow-up calls to caregivers associated with cats returned during the spring and summer months were postponed until autumn. Whenever possible, subsequent rounds of trapping were conducted by ACA at return sites where caregivers advised that additional cats were present.

### 2.2. Data Analysis

LMAS feline intake and euthanasia results for 2019 were compared to results from the baseline year of 2011. A similar process was employed to assess results for other shelter metrics, including live-release rate (LRR), adoptions, returned-to-owner (RTO), and length of stay (LOS). In addition, the number of sterilization surgeries performed as part of the LMAS RTF and the ACA TNR programs in the years following enactment of the 2012 community cat ordinance were examined. Comparisons to other communities where similar programs have been implemented, including on a normalized basis (i.e., per 1000 human residents), were also made. Due to fundamental differences in program implementation (e.g., TNR-only, RTF-only, and TNR and RTF combined), varied effort (e.g., the degree to which sterilizations were targeted in areas of greatest need), and inherent year-to-year variation in shelter metrics, no statistical analysis was attempted in comparing the LMAS/ACA program results to those documented in other communities.

## 3. Results

### 3.1. Enrollment and Surgeries

A combined total of 24,697 cats were sterilized in Jefferson County between 2012 and 2019 as part of the LMAS RTF (4989) and the ACA TNR program (19,708); an additional 9610 cats residing outside of Jefferson County were sterilized as part of the ACA TNR program. Cats 6 months of age or less made up 22.4% (6525/29,318) of the sterilization surgeries performed under the entirety of the ACA program (cats originating from both inside and outside of Jefferson County); such data were not tracked as part of the LMAS RTF program. Gravid females made up 7.3% (2135/29,318) of the cats sterilized as part of the ACA TNR program. Females in estrus comprised 5.3% (1544/29,318) of the total sterilizations, while males with cryptorchidism made up 0.5% (140/29,318) of the total number of cats sterilized; comparable data were not tracked as part of the LMAS program. Females made up 55.8% (17,270/30,936) and males 44.2% (13,666/30,936) of cats of known sex enrolled over the entirety of the ACA program.

Some of the cats enrolled in the ACA TNR program were already sterilized (1.7% female or 536/31,075, and 1.7% male or 519/31,075) or were not sterilized because they failed to meet minimum weight (2%, 609/31,075) or health (0.8%, 337/31,075) requirements upon enrollment, or because their surgeries were postponed by the veterinary staff for unspecified reasons (0.3%, 102/31,075). Although specific data were not available, ACA protocol was for all intact cats not sterilized immediately upon enrollment to be sterilized once they were deemed fit. Such cats were then either returned to locations of capture, transferred to KHS for adoption, or, at times, adopted directly by ACA staff. In addition, 47 cats (0.2%) died perioperatively.

Overall, from 2014 to 2019, 10,109 of the 19,147 community cats (52.8%) enrolled in Jefferson County, whether part of the LMAS or ACA programs, originated from the 10 “target” zip codes (Figure 3). Both LMAS and ACA sterilized cats from the target area at a disproportionally higher rate than the rest of the county (2180/3883, or 56.1%, and 7929/15,264, or 51.9%, respectively), as the zip codes comprising the target area make up 33% of the county’s human population and 30% of its land area [16]. An average of 6.5 community cats were sterilized per 1000 human residents in the target area as opposed to an average of 2.9 cats per 1000 residents in the remainder of the county between 2014 and 2019.

### 3.2. Impact on Shelter Metrics

At the end of 2019, feline euthanasia at LMAS had declined by 94.1% compared to a baseline of year-end 2011 (the year prior to initiation of the shelter’s RTF program), from 2626 to 155 cats (Table 1). The shelter’s feline euthanasia rate fell by 89.7% over the same eight-year period, from 55.6% to 5.7%. The shelter’s feline LRR increased by 147.6%, from 38% to 94.1%.

Annual feline intake at LMAS decreased 42.8% between 2011 and 2019, from 4727 to 2706 cats. Over that same period, stray feline intake decreased 28.0% (from 3341 to 2406 cats) and the intake of owner-surrendered cats decreased by 81.7% (from 1379 to 252 cats) (Figure 4).

Feline intake at LMAS dropped from 6.3 cats per 1000 human residents in Jefferson County in 2011 to 3.5 per 1000 residents in 2019; stray intake declined from 4.5 to 3.1 cats per 1000 residents over the same period (Figure 5). Euthanasia of cats decreased from 3.5 per 1000 human residents to 0.2 per 1000 residents.

In 2019, 468 cats trapped and sterilized as part of the ACA TNR program were counted in LMAS intake because the surgeries were paid for with municipal shelter funds (556 cats were handled in a similar fashion in 2018 and 870 in 2017); those cats were subtracted from LMAS intake for the purposes of this study since the cats in question were not actually physically admitted to the shelter and are accounted for in the ACA data.

Stray feline intake results by zip code were available only for the period of 2014–2019. During that period, 56.7% (7772/13,713) of stray intake originated from the 10 zip code target area. Over the six-year period for which data are available, stray intake at LMAS averaged 5.0 per 1000 human residents per year in the target area and 1.9 per 1000 residents annually for the remainder of the county. In 2014, the target area accounted for 60% (1419/2365) of stray intake at LMAS; by 2019, the percentage had fallen to 55.7% (1340/2406) (Figure 6). As a point of reference, stray intake at LMAS in the baseline year of 2011 was 3341 cats, but since no reliable data by zip code were available for years before 2014, the percentage originating from the target area at that time is unknown, but is thought to be at least what it was in 2014. This is based upon anecdotal accounts and the fact that the same parts of the county were the greatest sources of stray intake (see initial targeting project described in the Introduction) at a time when more stray cats were being admitted. From 2014 to 2019, stray intake in the target area declined by 5.6% (1419 to 1340); the same metric increased by 12.7% (946 to 1066) in the remainder of the county.

Compared to the baseline year of 2011, feline adoptions at LMAS increased by 20.6%, from 929 to 1120. As a percentage of total feline intake, adoptions rose from 19.7% to 41.4%. RTO increased from 2.2% to 3% of total feline intake (Table 1). Between 2011 and 2019, the average LOS for cats at LMAS declined by 61.1%, (from 17.5 days to 6.8 days); average LOS for cats admitted as strays declined 63.0%, from 17.0 to 6.3 days.

## 4. Discussion

The purpose of the present study was to examine changes in feline euthanasia, intake, and other metrics at the municipal animal shelter in Jefferson County, KY, USA, eight years after an RTF program was added to an ongoing community-based TNR program. A number of variables (e.g., program duration, program structure, available resources, climate, and differences in human and community cat population densities) make direct comparisons to other similar programs difficult. Nevertheless, the impact on feline euthanasia, intake, and other shelter metrics of the concurrent RTF and TNR programs in Jefferson County appear to be consistent with, or more favorable than, results associated with other programs; this is likely due to (1) the longer duration of the Jefferson County program, (2) sustained targeted TNR efforts over a large area encompassing the greatest sources of stray feline intake at the municipal shelter, and (3) the strong levels of collaboration between Alley Cat Advocates, a well-established and efficiently-run community-based TNR organization, and the municipal shelter.

### 4.1. Reductions in Euthanasia

Similar to what has been documented by other municipal shelters implementing RTF programs [11,14,27], a considerable reduction in feline euthanasia occurred at LMAS. The decline in feline euthanasia (94.1%) after eight years exceeded declines at the six municipal shelters with three-year CCPs (median of 83%) [11], and after 4 years in Jacksonville, FL, USA [14] and San José, CA, USA [27], where municipal shelters each experienced reductions of 67% after initiating RTF programs (Figure 7).

When considered on a normalized basis (i.e., per 1000 human residents), the decline in feline euthanasia experienced at LMAS (94.3%, from 3.5 to 0.2) also exceeded declines observed at the other locations (Table 2).

In addition, the feline euthanasia rate (calculated by dividing the number of cats euthanized for reasons other than owner request by the total number of feline intakes) decreased by 89.7%. This decline exceeded the median decline in euthanasia rate observed at the six shelters enrolled in the CCPs, which experienced a median reduction of 74% [11]. A euthanasia rate of 55.6% existed at LMAS in 2011, before implementation of their RTF program; the same metric for 2019 was 5.7%. By comparison, the six CCP-enrolled shelters had a median euthanasia rate of 36% prior to program initiation and a rate of 12% at program conclusion [11]. As an additional point of reference, Shelter Animals Count reported for 2018 an average feline euthanasia rate of 18.7% among its 775 participating organizations categorized as municipal shelters or organizations with municipal sheltering contracts [29].

The decline in overall feline intake documented by LMAS (42.8%) exceeded what occurred at the six shelters enrolled in the CCPs (median decline of 32%) [11] and in Jacksonville (30%) [14] and San José (27%) [27] (Figure 8). Although, the drop in overall feline intake (44%) was greater at the municipal shelter in Columbus (one of the CCP-enrolled shelters), where a much larger number of sterilization surgeries were performed per 1000 human residents (11.3) [11] than in Jefferson County. LMAS experienced the largest decline in overall feline intake on a normalized basis (6.3 to 3.5, or 44.4%) when compared to the other programs (Table 3).

### 4.2. Reductions in Stray Intake

The fact that LMAS documented a lower reduction in stray feline intake than in overall feline intake (28.0% and 42.8%, respectively) was unexpected; one might reasonably expect that, to the extent that a TNR program reduces feline intake, the impact would be seen mostly in the admission of cats deemed “strays.” However, over the 8 years documented here, LMAS also observed a decline in owner-surrendered cats of 81.7%, from 1379 to 252 cats. This reduction of 1127 cats made up 55.8% of the reduction in overall feline intake. Although data to determine the exact cause(s) for this are unavailable, it is likely that the following three factors contributed strongly. First, the increased availability of sterilization (through ACA and, to a lesser degree, LMAS) almost certainly led to an increase in the sterilization of pet cats as well. Many of these cats would not have been identified as such by either ACA or LMAS, and would have been sterilized as if they were community cats. By reducing the number of kittens born to these cats, the programs reduced the admission of kittens in the future. Indeed, the three-year CCPs described previously documented a median decline of 40% in the admission of kittens across the six participating shelters, and a 41% median decline in the admission of newborn kittens at the four shelters that tracked such data [11]. Secondly, KHS opened a low-cost clinic in 2007, adding considerably to the local surgical capacity available to pet owners and colony caregivers. As with the ACA and LMAS efforts, this likely led to fewer kitten births and thereby reduced feline intake. Finally, LMAS began implementing its managed-intake process in 2017, whereby the intake of owner-surrendered animals is scheduled by appointment in order to match intake with available space and resources.

### 4.3. Other Shelter Metrics of Interest

Although reductions in average LOS are sometimes considered inherently positive, such determinations must also consider eventual outcomes. Euthanizing a large number of cats immediately following the expiration of a legally mandated stray-hold period, for example, would likely produce a reduction in average LOS but few would consider this positive overall. As Figure 9 illustrates, low euthanasia rates can be achieved despite a relatively high average LOS. Moreover, factors increasing LOS need not have a negative impact on euthanasia. The data depicted in Figure 9 reveal three general groupings: (1) 2011–2013, when the RTF program was begun, thought to be responsible for the largest year-over-year reduction in euthanasia; (2) 2014–2016, when medical care of community cats was transferred from ACA to LMAS, thereby increasing LOS; and (3) 2017–2019, when a second surgery suite was added, thereby shortening wait times for sterilizations and other medical procedures.

The increase in LRR, 147.6% (38% to 94.1%), experienced at LMAS exceeded the median change in LRR observed at the six shelters enrolled in the CCPs (53%, median LRR of 57% before the program and 83% afterward) [11]; the post-program LRR at LMAS also surpassed the LRR (76.7%) among municipal shelters and organizations with municipal sheltering contracts as reported for 2018 by Shelter Animals Count [29]. Post-program RTO at LMAS (3%) exceeded the median for the same metric at the six CCP shelters (2%) and was consistent with the RTO for similar facilities (3.3%) reported for 2018 by Shelter Animals Count [11,29].

The increase in the absolute number of adoptions at LMAS (20.6%) as well as the rise in adoptions as a percentage of feline intake (19.7% to 41.4%) are thought to be the result of both lower total feline intake associated with the implementation of RTF in 2012 and the introduction of a managed-intake approach for owner-surrendered cats at LMAS in 2017. The same two factors are credited with the reduction of 61.1% in the average LOS (17.5 days to 6.8 days) for cats at LMAS.

Similar to what occurred at the six shelters enrolled in the three-year CCPs [11], the number of cats sterilized as part of ACA’s community-based TNR efforts in Jefferson County (between 2012 and 2019) outnumbered those sterilized as part of the RTF program at LMAS by a ratio of 4:1 (19,708 to 4989). Overall, during the years for which data are available (2014 to 2019), the ratio of cats trapped and returned to the target area as part of the respective TNR and RTF efforts also approached 4:1 (7929 to 2180).

The cats enrolled in the ACA TNR program were generally in good health, as is illustrated by the low incidence of cats dying in care, which is consistent with what has been observed at other locations where similar TNR programs have been implemented [11,30]. In fact, a growing body of evidence suggests that cats enrolled in TNR programs are generally free of grave illness or injury: among more than 104,000 cats enrolled in ACA’s TNR program and enrolled as part of the six CCPs, 0.2% died in care [11]; similarly, 0.3% of enrolled cats died in care over the course of two programs, totaling 7689 cats, conducted at different times in Alachua County, Florida [26,30].

### 4.4. Key Differences in Jefferson County’s Trap-Neuter-Return (TNR) and Return-to-Field (RTF) Programs

Although similar in many respects to other programs that employed community-based targeted TNR and/or shelter-based RTF, the Jefferson County program differed in important ways. Unlike the CCPs, that employed a tactic known as the red-flag cat model whereby additional cats identified as living near RTF program return sites were the focus of subsequent TNR program targeted trapping efforts [11], ACA staff consulted with colony caregivers (as described above) in order to trap, sterilize, and return additional cats as part of their community-based TNR program; LMAS conducted no similar trapping program.

In addition, unlike a high-intensity targeted TNR program in Alachua County that achieved large declines in both feline intake and euthanasia by focusing efforts on a single zip code for a relatively short period of time (2 years) [30], the targeted TNR component of the Jefferson County program was sustained for eight years and focused on a large geographic area (30% of the county land area) that is home to a significant portion of the county’s residents (33%). The target area in Jefferson County was identified as a disproportionately large source of feline intake at LMAS (~60% in 2011). In Alachua County, by contrast, the target area covered 0.5% of the county land area and was comprised of 7.5% of the human population; it had the highest shelter intake of any zip code in the county (6.2% of the municipal shelter’s total feline intake). In Alachua County, an average of 61 cats each year per 1000 human residents were sterilized in the target area and 12 cats annually per 1000 residents in the remainder of the county. Much larger declines (on a normalized basis) in feline euthanasia (95%) and intake (69%) were observed in the target area than the remainder of the county (13% and 30%, respectively) [30]. In Jefferson County, no sterilization or intake information specific to the target area was available for the baseline year or the first two years of the program, so the impact on shelter metrics in the target area could not be calculated (but because the number of cats sterilized per 1000 residents across the county as a whole—approximately 4 annually—was much smaller than in Alachua County the impact on shelter metrics relating to the target area was likely much less). Notably, however, reductions in overall feline intake (i.e., county-wide) at LMAS at year-end 2014 (two-plus years after program inception in mid-2012) were double those observed at Alachua County Animal Services after two years: from 4727 to 3226 (31.8%) cats at LMAS, and from 3996 to 3398 (15%) at the Alachua County shelter. The difference is likely attributable to the pairing of RTF with community-based targeted TNR in Jefferson County; a similar decline (median of 32%) in feline intake was observed over 3 years among the six CCP shelters [11]. The fact that between 2014 and 2019 stray intake from the targeted zip codes within Jefferson County declined by a comparatively smaller percentage (as described above) was unexpected, but is likely due to the diluted intensity of the targeting program as well as an increased use of the municipal shelter as a resource for companion animals, including community cats, due at least in part to new perceptions among residents of the shelter as a “cat-friendly” facility after commencement of the RTF program.

The results of this study provide compelling evidence that combining RTF with targeted TNR, even when targeting is performed at a lower intensity, is generally more effective than targeted TNR alone at reducing feline intake over a shelter’s entire service area. In addition, the intensity of the targeted TNR efforts necessary to achieve similar results across different communities is very likely to be a function of a number of key factors (e.g., the existing levels of community cat sterilizations prior to implementation of RTF/TNR, existing cat populations and densities, and resource availability) rather than a universal standard.

## 5. Study Limitations

As this was a retrospective investigation, it was limited by the available data. For instance, kitten-specific shelter results were not tracked, and results by zip code were not available from LMAS for years prior to 2014. Data related to the characteristics of cats enrolled in the RTF program, such as age and sex, were not available from LMAS. Some information (e.g., sex and pregnancy status) for some years was tracked by ACA only on a program-wide basis, not by county or by zip code. In addition, no tracking was performed of the additional cats sterilized based upon caregiver input as part of ACA’s targeted TNR program, nor was data related to colony size collected. Finally, the welfare outcomes for cats returned to sites of origination were not monitored by the RTF or the TNR program, precluding analysis.

## 6. Conclusions

A growing number of municipal shelters, located in communities of different sizes across the USA, have experienced improved metrics, particularly reductions in feline intake and euthanasia, after the implementation of RTF and targeted TNR programs. The coexistent programs in Jefferson County demonstrate the adaptability of these two community cat management tactics in terms of tailoring their implementation to meet the specific needs and resources of a given community.

Examples of such adaptability include the following alternative methods of implementation: initiating an integrated shelter-led program that combines RTF and targeted TNR (as was carried out in the case of the CCPs) or conducting separate but coordinated public and private programs (as was carried out in Jefferson County); use of the red-flag cat model to sterilize additional cats residing near RTF return sites (as was carried out as part of the CCPs) or employment of a modified version of the same model, but rather as an extension of community-based TNR efforts that utilize the input of colony caregivers in order to identify additional cats to sterilize at targeted locations (as was carried out in Jefferson County); utilization of high-intensity short-term targeting (as was carried out in Alachua County) or use of sustained proportional targeting (as carried out in Jefferson County). There is significant evidence that all such variations of these community cat management tactics have contributed to declines in feline intake and euthanasia at municipal shelters.

## Figures and Tables

**Figure 1 animals-10-01395-f001:**
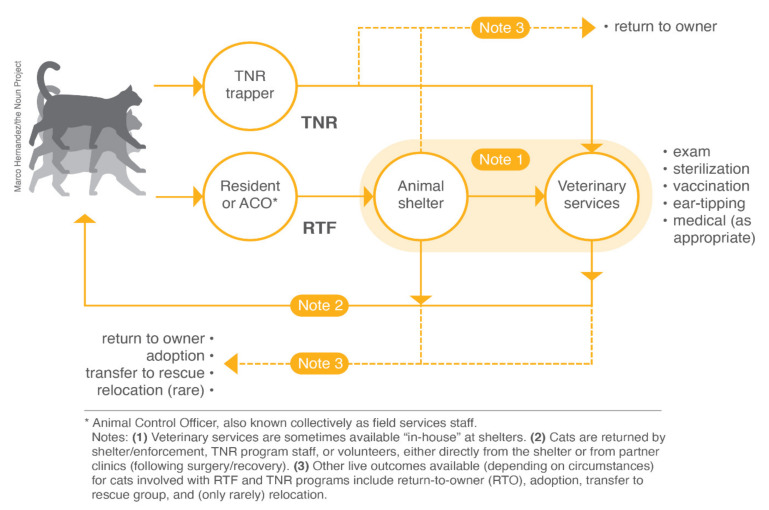
Visual representations of trap-neuter-return (TNR) and return-to-field (RTF) programs, adapted from [11].

**Figure 2 animals-10-01395-f002:**
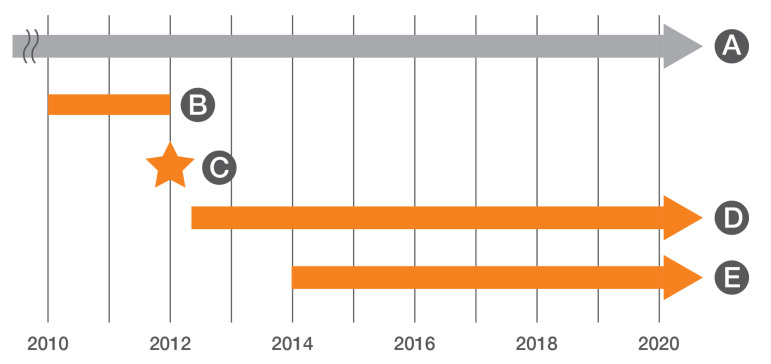
Timeline of key events reported in the present study. A: Alley Cat Advocates (ACA)’s trap-neuter-return (TNR) county-wide program (beginning in 1999); B: ACA’s targeted TNR in zip code 40215; C: Louisville Metro, KY, USA, approves TNR ordinance; D: LMAS implements return-to-field (RTF) program for service area; E: ACA focuses TNR efforts on 10 zip code target area.

**Figure 3 animals-10-01395-f003:**
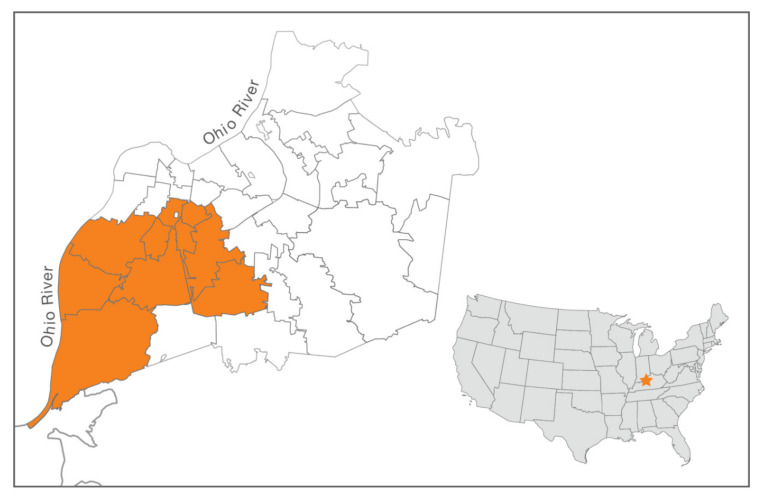
Map of Jefferson County, KY, USA, showing target (orange) and non-target (white) areas corresponding to different sterilization efforts.

**Figure 4 animals-10-01395-f004:**
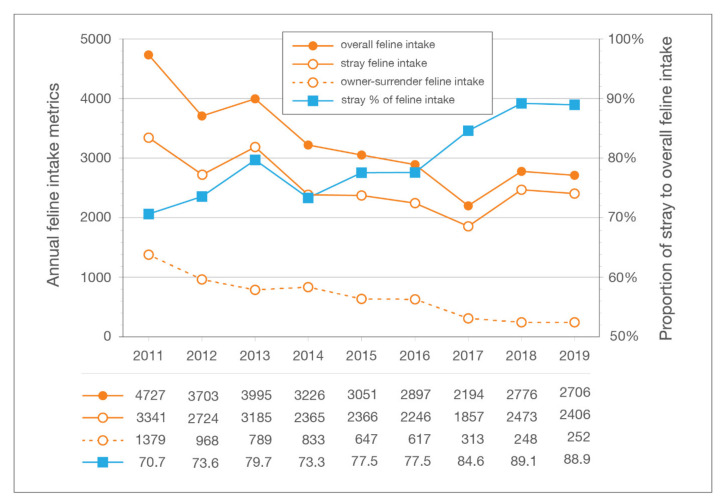
Overall, stray, and owner-surrendered feline intake for each year of the LMAS/ACA program. LMAS: Louisville Metro Animal Services; ACA: Alley Cat Advocates.

**Figure 5 animals-10-01395-f005:**
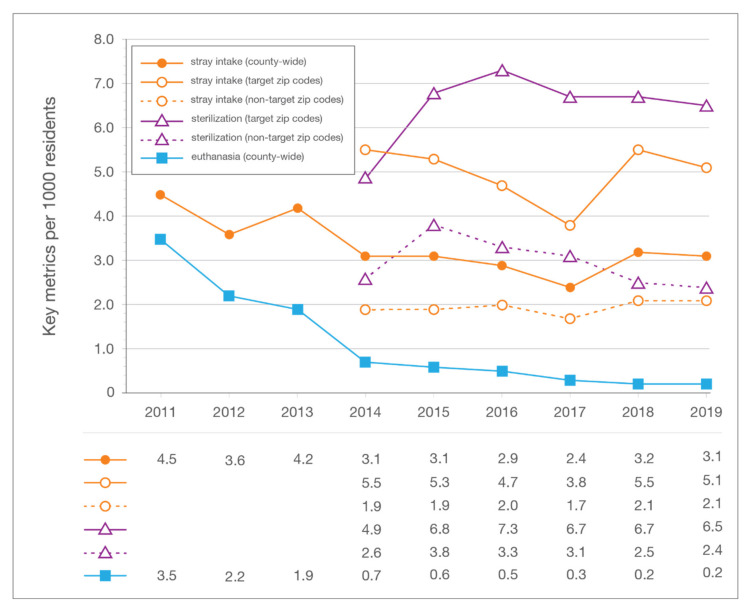
In-county (target and non-target zip code area) sterilization surgeries, stray feline intake, and feline euthanasia per 1000 residents for each year of the LMAS/ACA program. LMAS: Louisville Metro Animal Services; ACA: Alley Cat Advocates.

**Figure 6 animals-10-01395-f006:**
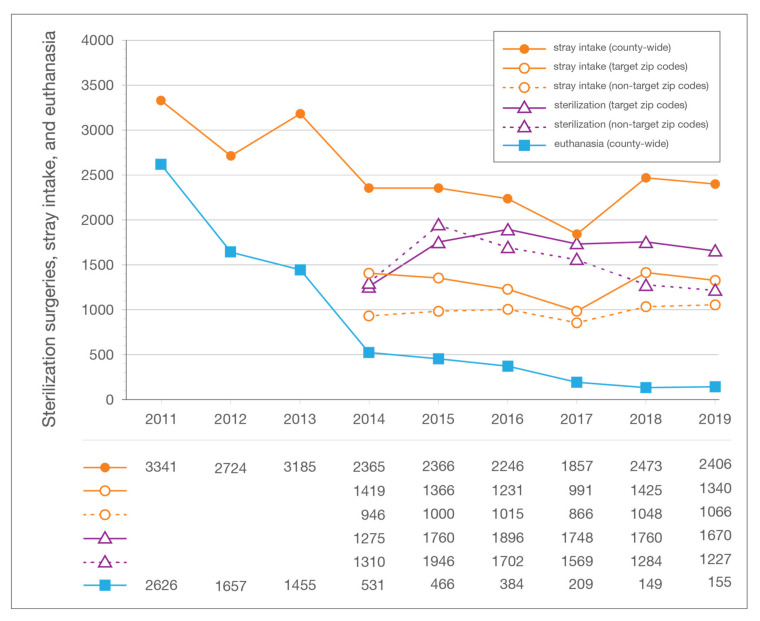
In-county (target and non-target zip code area) sterilization surgeries, stray feline intake, and feline euthanasia for each year of the LMAS/ACA program. LMAS: Louisville Metro Animal Services; ACA: Alley Cat Advocates.

**Figure 7 animals-10-01395-f007:**
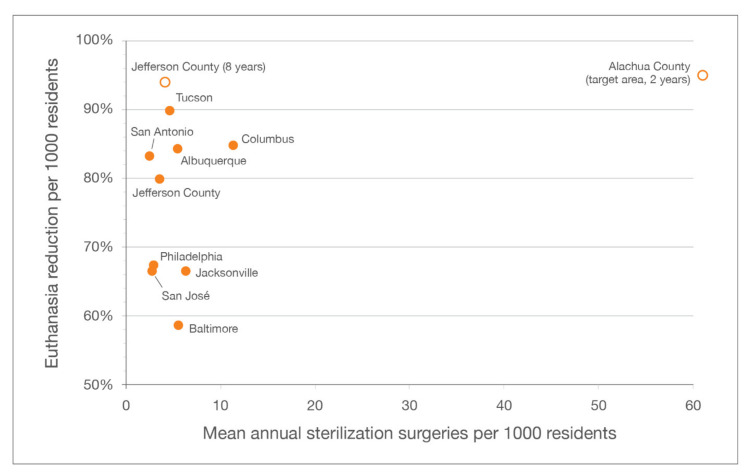
Reductions in feline euthanasia per 1000 residents over three years, across different communities and programs.

**Figure 8 animals-10-01395-f008:**
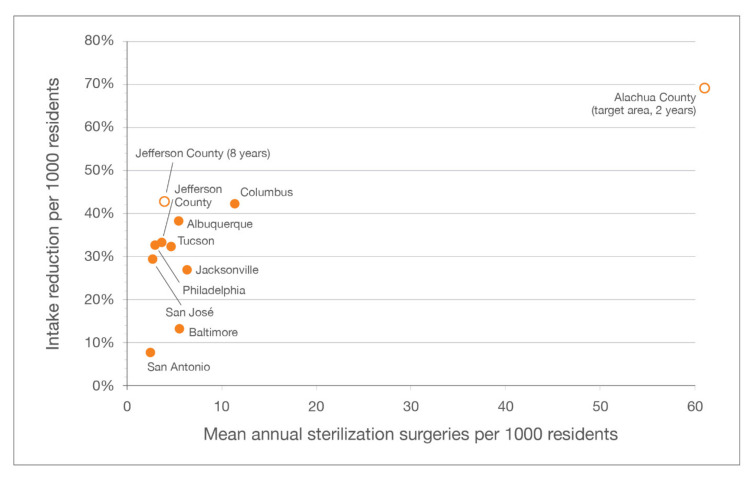
Reductions in feline intake per 1000 residents over three years, across different communities and programs.

**Figure 9 animals-10-01395-f009:**
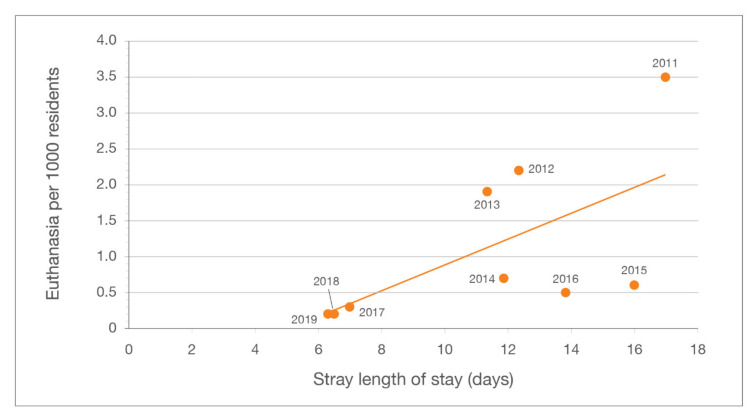
Feline euthanasia per 1000 residents vs. average length of stay (LOS) for cats admitted as strays.

**Table 1 animals-10-01395-t001:** Summary of key Louisville Metro Animal Services (LMAS) metrics by intake and outcome category, 2011–2019.

	2011 (Baseline)	Y1	Y2	Y3	Y4	Y5	Y6	Y7	Y8
Intake Category									
Stray intake	3341	2724	3185	2365	2366	2246	1857	2473	2406
Owner surrendered	1379	968	789	833	647	617	313	248	252
Confiscation	7	11	21	28	38	34	24	55	48
Total feline intake	4727	3703	3995	3226	3051	2897	2194	2776	2706
**Outcome category**									
Returned *	0	373	891	627	811	793	262	996	1080
RTF	0	313	793	500	724	701	131	907	920
Medical only	0	60	98	127	87	92	131	89	160
% of strays returned	0.0	13.7	28.0	26.5	34.3	35.3	14.1	40.3	44.9
Adoptions	929	884	840	899	890	1043	1166	1036	1120
To rescue †	661	595	704	1008	624	562	572	427	191
RTO	75	95	72	62	55	51	61	73	87
Euthanasia	2626	1657	1455	531	466	384	209	149	155
Died in care	144	92	60	45	45	72	48	33	54
Feline LRR (%)	38.0	53.7	63.0	83.2	83.8	86.7	91.7	94.4	94.1

* Includes cats returned as part of the LMAS RTF program as well as those already sterilized and later brought to LMAS for medical services. † Like many municipal shelters, LMAS partners with local rescue groups, which facilitate the adoption of the animals in their care; Y: Year; RTF: Return-to-Field; RTO: Return-to Owner; LRR: Live-release Rate.

**Table 2 animals-10-01395-t002:** Impact of LMAS RTF and ACA TNR programs in terms of feline euthanasia per 1000 human residents, and comparison to similar programs in other communities. Sources of data: Jacksonville, FL [14,28]; San José, CA [27]; Albuquerque, NM, Baltimore, MD, Columbus, GA, Philadelphia, PA, San Antonio, TX, and Tucson, AZ [11].

Community/Program	Baseline	Y1	Y2	Y3	Y4	Y5	Y6	Y7	Y8	Mean Annual Sterilizations
Jefferson County, KY	3.5	2.2	1.9	0.7	0.6	0.5	0.3	0.2	0.2	4.0
Jacksonville, FL	13.2	11.3	7.0	4.4	3.8	2.5				6.0
San José, CA	7.2	5.5	3.2	2.4	2.2	1.6				2.7
Albuquerque, NM	4.5	1.9	1.0	0.7						5.4
Baltimore, MD	3.4	1.7	1.7	1.4						5.5
Columbus, GA	7.3	2.9	1.2	1.1						11.3
Philadelphia, PA	4.0	2.5	1.8	1.3						2.9
San Antonio, TX	2.4	1.7	0.7	0.4						2.4
Tucson, AZ	3.0	0.4	0.3	0.3						4.6

LMAS: Louisville Metro Animal Services; RTF: Return-to Field; ACA: Alley Cat Advocates; TNR: Trap-Neuter-Return; Y: Year.

**Table 3 animals-10-01395-t003:** Impact of LMAS RTF and ACA TNR programs in terms of feline intake per 1000 human residents, and comparison to similar programs in other communities. Sources of data: Jacksonville, FL [14,28]; San José, CA [27]; Albuquerque, NM, Baltimore, MD, Columbus, GA, Philadelphia, PA, San Antonio, TX, and Tucson, AZ [11].

Community/Program	Baseline	Y1	Y2	Y3	Y4	Y5	Y6	Y7	Y8	Mean Annual Sterilizations
Jefferson County, KY	6.3	4.9	5.3	4.2	4.0	3.8	2.8	3.6	3.5	4.0
Jacksonville, FL	15.6	15.0	14.6	11.4	10.9	10.5				6.0
San José, CA	10.2	9.1	8.2	7.2	7.6	7.0				2.7
Albuquerque, NM	14.6	11.9	9.8	9.0						5.4
Baltimore, MD	11.2	10.8	9.6	9.7						5.5
Columbus, GA	16.3	12.1	9.7	9.4						11.3
Philadelphia, PA	12.2	10.8	9.0	8.2						2.9
San Antonio, TX	3.8	5.6	3.6	3.5						2.4
Tucson, AZ	7.7	6.0	5.9	5.2						4.6

LMAS: Louisville Metro Animal Services; RTF: Return-to Field; ACA: Alley Cat Advocates; TNR: Trap-Neuter-Return; Y: Year.

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
