# Peer review of "The Impact of Return-to-Field and Targeted Trap-Neuter-Return on Feline Intake and Euthanasia at a Municipal Animal Shelter in Jefferson County, Kentucky"

_animals, 2020, doi:10.3390/ani10081395_

Round 1
Reviewer 1 Report
The authors have done a good job of addressing the specifics I have raised. The paper is well written and thorough.
My fundamental issues however still remain
a) I believe there is a lot of novel information presented in the discussion which belongs in the results section
b) This paper is essentially a programme description. Although metrics are rigorously collected documenting the success of the programme, there is no central research question. Therefore I question the novelty of the paper and to what extent it belongs in the scientific literature as compared to a book describing similar approaches and programmes. The addition of a few sentences has endeavoured to explore the reasons for the apparent high success of the programme but it remains de primarily descriptive.
Reviewer 2 Report
I appreciate the considerations the authors have given to my suggestions and accept their explanations (maybe a future paper...)
I found only one more minor mistake:
L14: management OF ....
This manuscript is a resubmission of an earlier submission. The following is a list of the peer review reports and author responses from that submission.
Round 1
Reviewer 1 Report
This paper describes an impressive case study of the results of a TNR Program in Jefferson County USA. It is an extremely thorough account of the invention and includes much descriptive analysis of potential outcome measures, but the paper currently lacks a central research question and demonstration of any novel contribution to the field.
The structure of the paper is currently inconsistent; with aspects of the introduction appearing in the methods section and results in the discussion. I believe the material is essentially a programme description and hence its publication belongs in a dedicated text book or on a website, where similar programmes are described, rather than in a scientific peer reviewed journal where this field now benefits form novel ideas and syntheses such as that by Foley et al (2007: https://www.avma.org/sites/default/files/resources/javma_227_11_1775.pdf)
Specific comments
There is a lot of terminology, jargon and acronyms, Whilst the authors do a good job of defining these at the outset, the paper remains really heavy work for a no- specialist to read, which limits its readability.
The Simple summary would benefit from some description of what Return to Field entails, currently this section reads very similarly to the scientific abstract and would benefit from more description for the lay person
Line 71 “the Jefferson County programs were operated on a concurrent, yet independent basis—the RTF program operated by municipal shelter staff and the TNR program, including the sustained targeting of specific zip codes, conducted by a private community cat advocacy organization.” This sentence is unclear to me. Please can you explain this difference and its relevance more clearly.as well as how you would predict this difference may impact upon your outcome measures.
102 Is Live-release Rate (LRR) a ratio or a no per year? To standardise with other definitions please state units.
111-156 describes the background that drove the TNR initiative, I I think this belongs in the introduction not the methods section
The meaning of “Returns” in table 1 is not clear
Fig 3 does not make sense to me, how can the stray % intake in take increase in spite of the number of stray feline intake numbers going down
Fig 4 the meaning of target and non-target is not clear in this diagram
Table 2 and much of the information presented in this section belongs in the results not the discussion
It’s not clear what the numbers in Table 3 represent, and I do not follow the sentence “considered on a normalized basis, the decline in feline euthanasia” .. what does this mean?
Figure 7 shows limited data which may be better shown in a table
The discussion in general remains very descriptive. I feel the key point here needs to be explaining why the outcomes from this programme compare so favourably with others……What are the key drivers for this? This is mentioned briefly at the end of the discussion but I think needs to be the main research question and focus of any scientific paper based on this data.
Reviewer 2 Report
The study by Spehar and Wolf provides important information on the success of Trap-Neuter-Return studies in terms of shelter intake rates. (It is good to see that the scheme seems to have been very successful in the aspects investigated here). At the moment, while I think the study is worthwhile publishing, there are still a couple of issues with it. Most importantly, it is currently descriptive, while I think it could be strongly enhanced by conducting a couple of simple statistical tests (nothing more complicated than correlations or paired tests needed). To some extent in connection with this, I feel that at the moment the structure is not quite optimal, because quite a lot of the “descriptive analysis” is done in the Discussion, because the authors refer to other studies and set the finding from this particular programme into the context of these other studies. However, if this would be turned into a proper analysis (based on published data and the data from this specific programme), all this could go into the Results were it would be better suited, and the Discussion could concentrate on the deeper evaluation of the findings.
I have also a bit of a problem with fully understanding the terms TNR vs. RTF (see comments below), as well as the use of the term “sterilisation” vs. “neutering” vs. “spaying” (again, see specific comments).
As a general point: Do you have any numbers on how many feral plus stray cats were living in the relevant areas over the years? If you have the data and could include it (or if that is in preparation refer to it) that would be great. There are people that deny the usefulness of TNR programmes, and this would be an important piece of information in that context. (Your Limitations section seems to suggest that you don’t have these data – what a shame!)
Terminology, clearer Methods; separate Methods from Results.
L 56-58: In the Abstract (and given that two different terms and acronyms are used) it sounds as if TNR and RTF approaches are different. The sentence here suggest that they mean exactly the same, i.e. that in both cats are, for example, neutered before return. When reading only the abstract I had been wondering whether RTF perhaps includes returning without neutering. Maybe it should be stated yet more explicitly (perhaps also in the abstract) that the two terms are used synonymously. If they are not synonymous, please explain the difference more clearly. L 68 again suggests that the two are somewhat different (otherwise you could not “pair” them). The following lines suggest that the two terms rather reflect WHO is doing the programme, and not really what is done. It would be good to make that clearer earlier on. Possibly simply refer to your “key terminology” further below.
L60/L86: As far as I know the term “sterilized” has a slightly different meaning than “castrate” or “neuter” where sterilization does not completely remove the organs (ovaries or testicles), while castration/neutering does. [I am non-native speaker – and that might only be the usage in my language.] Please use the accurate term, and do not use them interchangeable. Perhaps rather use “neutering” throughout?
L119: Please provide (also) the metric dimensions.
L137: For better understanding, perhaps add “area” (or something like that) before the ZIP code.
L160: As the term “neutering” is used for both males and females the “spay or” is unnecessary.
L218: It seems that the term neutering is used here only for males. Since this is not the general definition (at least not in all English speaking countries: in the UK, for example, neutering is definitely used for both males and females), you should define how you use the terms or better use the generally used terms to avoid confusion.
Methods/Section 2.1: Due to the lengthy explanation of what happened BEFORE this study started, it becomes a bit confusing what was exactly done for this study. For example,
L233/234: Here you mention target zip codes for the first time. This should be explained in the Methods.
Also, you do not seem to refer to Fig. 2 and some of the other figures in the text.
L235: Sorry, I am getting lost here. What is the denominator here (the 3881 and 15,264)?
L236: Are you talking about the human or the cat population here?
L237: Just to make it clearer: Are you here talking about community cats or cats in general?
L243 following: These follow up calls should be mentioned in the methods.
L255: With “overall euthanazia rate” do you mean in the county as a whole? Perhaps state that more clearly. A reader not familiar with the area and the whole circumstances/methodological approach may get a bit lost in places what your comparisons are in each case.
Table 1: Perhaps state explicitly in the table header that “baseline” refers to the year 2011.
Also, what does the “to rescue” outcome refer to? The term seems nowhere to be explained.
Also, You refer to “stray intake” – is that really only strays as defined in section 1.1 (i.e. previously owned cats that got lost or were abandoned) or do you mean all types of community cats? – same for Fig. 3 & 4. (also in the text, e.g. paragraph starting L 276)
Fig. 3: The numbers in the figure label are a bit irritating and not necessary for the figure. Furthermore, the values are all provided in Table 1. Remove the grid lines from the figure.
Fig. 4: With “surgeries”, do you mean any type of surgeries or specifically neutering? In that case, better use the term neutering to avoid confusion.
Similar to Fig 3 I don’t really like the numbers below the figure 4 & 5– but in this case no table is available. One part of me would say that Fig. 3 and Table 1 are identical, so only one is needed, but I think both are useful (Figures for the overview, tables for the details). I guess that was the intention of having the numbers under the Figure. But I think it would still look better separately. Maybe all values could also be added to Table 1.
L283: Based on what evidence did you think that the percentage was at least what it was in 2014?
L294/5: Table 2 should already be referred to first in the Results section, not in the Discussion.
Table 2: What do you mean by “medical only”?
Generally, in the Discussion there is quite a bit of repetition of Results, and some aspects have not been mentioned before in the Results (e.g. around euthanasia). Try to separate the two sections more clearly. I think part of the problem is that you review also findings from other locations in Tables 2 and 3 – this “feels” more like a Result. (I agree it is somewhat borderline.) Perhaps you could consider stating as part of the Methodology that you also review findings from similar trials elsewhere and then present these (Table 2, 3, Fig. 6 etc.) in the Results. It is the fact that there are not previously reported values/numbers in the Discussion that I find confusing. .
Maybe, you might even do a more quantitative approach for some of these findings. For example you might consider a (probably non-parametric) correlation analysis of annual neuterings vs.intake reduction (Fig. 7), or, if you have before-after data for each of these programmes, a paired t-test or paired Wilcoxon signed rank test would be even better (and the same for data in Fig. 6; Fig. 8 prob. best as correlation, e.g. Spearman’s). This quantitative approach would greatly enhance the strength of your study that, as it is, is “only” descriptive.*
* I did a quick test on the data in Table 3 and as expected, you have a nice statistically significant result: Wilcoxon signed rank test: V=45, df=8, p=0.009.
As I note that at least one of the authors is not at a research institution and I do not know how comfortable they are with doing statistical analyses, I am happy to offer, that if the authors need help with statistical analyses, they can contact the editor to reveal my details and I would be happy to help. (Please don’t take offence by this offer if you are not in need of help, it’s not meant condescendingly in any way.)
Fig. 6: specify “over three years – from when to when?
L342: replace “it’s” with “it is”
L343: replace the word “significantly” (since you have not conducted a statistical analysis) with “strongly” or something similar.7
L432/433: This is again about the confusing distinction between TNR and RTF. After reading section 1.1 I thought the difference was mainly about WHO did it, but this sentence again suggests a more fundamental difference. Maybe if what is meant that shelters or communities on their own do not quite as well on their own than if they combine their efforts then write it like this (in words) rather than contrasting the terms.
Round 2
Reviewer 2 Report
I am mostly happy with the responses of the authors where they did not follow my suggestions.
I am still not 100% convinced that some statistical analyses would not be possible. I appreciate the concern that statistical tests should not be conducted if they are not appropriate, and being not familiar with the intricacies of the data-set makes it obviously difficult to judge this. You are probably right that for simple correlations the data are too varied. But I think that the argument that the underlying differences between shelters necessarily render comparisons invalid is not necessarily true. Firstly, if input data are very different this would rather tend to obscure significant (both in the statistical and in the general sense) relationships, i.e., it would make it more likely to find statistically significant relationships if in fact there is an underlying effect. The only exception would be if there is an inherent bias. Secondly, if paired-tests are a possibility (same shelter at baseline year and at the final year of programme or Year three), would take these inherent differences into account, unless the approaches changed within the same shelter or community. In fact, that would be exactly the reason why a paired test would be used – and its strength. To some extent the argument by the authors also does not seem to be very convincing because if qualitatively these comparisons can be made (as they do), then it should also be possible to do it quantitatively, as the arguments about different approaches would also hold for qualitative comparisons. So phrases like the one L336-340 do effectively make the statement that there is a general pattern, but the authors argue in their response to my comment that statistically it cannot be tested because they are not comparable.
So what I am saying is: you may not be able to do stats on your own Jefferson data set, but you could do stats using all the different studies (despite their different approaches) to make the general statement that TNR/RTF programmes are successfully reducing intake rates, euthanasia rates etc., regardless of the specifics of the approach (or rather taking these differences into account by using a paired test). Obviously, that doesn’t solve the question whether there are yet additional factors that in general these values declined (something that would definitely have to be raised in the Discussion), which would only be possible to answer if other shelters/communities were included that don’t have a TNR/RTF approach – but that is the same problem whether you do the stats or not.
However, if the authors, who know their data set better, are still convinced that this would not be appropriate, I do not press this point.
Only a few minor comments left:
L78: The brackets in the middle of the sentence disrupt the flow .and ease of understanding of the sentence. Consider rephrasing (making the content in the brackets into a separate sentence), or placing the bracket at the end of the clause, rather than in the middle of it.
L101: Remove full stop before bracket referring to Fig. 1.
L104-106: Thanks for the clarification.
Fig. 1: Explain the acronym ACO within the figure label.
L337, L385: If you do not do statistical tests, do not use the term “significant”, but rather. I think you CAN make the statement – but only if you do the paired-tests I suggested above.